# Effects of a modified muscle sparing posterior technique in hip hemiarthroplasty for displaced intracapsular fractures on postoperative function compared to a standard lateral approach (HemiSPAIRE): protocol for a randomised controlled trial

Anna Price ,[1] Susan Ball ,[2] Shelley Rhodes,[1] Robert Wickins,[3] Elizabeth Gordon,[4] Alex Aylward,[5] Emma Cockcroft,[2] Sarah Morgan-Trimmer ,[6] Roy Powell,[7] John Timperley,[8] John Charity[8]

For numbered affiliations see end of article.

**Correspondence to**
Dr Anna Price;
a.price@exeter.ac.uk

## ABSTRACT

**Introduction** Currently National Institute for Health and Care Excellence clinical guidelines in the UK suggest that surgeons performing partial hip replacements (hemiarthroplasty) should consider using the lateral approach. Alternatively, a newer, modified posterior approach using a muscle sparing technique named 'Save Piriformis and Internus, Repairing Externus' (SPAIRE) can be used leaving the major muscles intact. This randomised controlled trial (RCT) aims to compare the SPAIRE approach to the standard lateral approach, to determine if it allows patients to mobilise better and experience improved function after surgery.

**Methods and analysis** HemiSPAIRE is a two-arm, assessor-blinded, definitive pragmatic RCT with nested pilot and qualitative studies. Two hundred and twenty-eight participants with displaced intracapsular fractures requiring hip hemiarthroplasty will be individually randomised 1:1 to either the SPAIRE, or control (standard lateral approach) surgical procedure. Outcomes will be assessed at postoperative day 3 (POD3) and 120 (POD120). The primary outcome measure will be level of function and mobility using the Oxford Hip Score at POD120. Secondary outcomes include: De Morton Mobility Index (DEMMI), Cumulated Ambulatory Score and Numeric Pain Rating Scale (NPRS) at POD3; DEMMI, NPRS and EQ-5D-5L at POD120, complications, acute and total length of hospital stay, and mortality. Primary analysis will be on an intention-to-treat basis. Participant experiences of the impact of surgery and recovery period will be examined via up to 20 semi-structured telephone interviews.

**Ethics and dissemination** The protocol has been approved by Yorkshire and the Humber—Bradford Leeds Research Ethics Committee. Recruitment commenced in November 2019. Findings will be disseminated via research articles in peer-reviewed journals, presentations at conferences, public involvement events, patient groups

## Strengths and limitations of this study

► Comprehensive patient and public involvement: outcome measures were inspired by patients who said that mobility and speed to regaining independence are the most important outcomes after partial hip replacement surgery.

► Addresses the current evidence gap on the impact on patient mobilisation and function after surgery using different techniques (Save Piriformis and Internus, Repairing Externus vs lateral).

► A pragmatic multicentre study across six National Health Service Trusts in the South West of England with broad inclusion criteria to recruit a population that will mirror as close as possible this frail group of patients as reduced cognition is not an exclusion criterion.

► Due to the frail characteristics of the target population, a relatively high dropout rate is anticipated, however, this has been accounted for in the sample size calculation.

► Due to COVID-19, the primary outcome measure was changed to a self-report measure, meaning some outcomes were not collected in the manner originally planned.

and media releases. A summary of the trial findings will be shared with participants at the end of the study.
**Trial registration number** NCT04095611.

## INTRODUCTION

Hip fractures are common in the elderly. Over 20 000 cases of hip hemiarthroplasty (replacement of the fractured femoral head) are performed annually in England, Wales and



Northern Ireland.[1] Hip fracture patients endure debilitating loss of function, and recovery is often complex and challenging.[2] The average total length of stay for hip fracture admissions is over 21 days, representing over 4000 National Health Service (NHS) hospital beds occupied by hip fracture patients at any one time.[3]

The National Institute for Health and Care Excellence (NICE) recommends replacement arthroplasty (total hip replacement or hemiarthroplasty) to patients with a displaced intracapsular hip fracture. For patients who are not eligible for total hip replacement, hemiarthroplasty should be offered.[4] As a commonly performed procedure, it is important to consider innovative hemiarthroplasty techniques that may allow better and safer rehabilitation in this frail group of patients.

When planning a hip hemiarthroplasty, surgeons have a choice of surgical approaches. NICE guidelines[4] currently recommend using a lateral rather than a conventional posterior approach. This advice is based on evidence described as being of 'very poor quality'.[4] For adequate exposure, the lateral approach requires division and subsequent repair of 50% or more of the tendon attachments of the gluteus medius and minimus muscles on to the greater trochanter. These muscles are essential for normal gait. Literature quoted suggests a reduced dislocation rate for the lateral approach; however, this surgery has the disadvantage that the relatively extensive division of tendon attachments required may result in reduced levels of function postoperatively. Recent evidence from a cohort of over 20 000 patients from the Norwegian Hip Fracture Register[5] reported better patient-related outcome measures (pain, patient satisfaction and health-related quality of life) with a standard posterior approach compared with the lateral approach. However, the study also confirmed the higher dislocation rates in the conventional posterior approach group. Some studies report dislocation in up to 10% of patients undergoing this hip procedure through a standard posterior approach and such complications can lead to potentially catastrophic consequences.[6]

To address the issue of dislocation in hip arthroplasty, modified surgical procedures have been attempted using muscle sparing techniques. In 2012, Han et al[6] described a modified posterior approach for use in patients with neurological disorders requiring hip hemiarthroplasty, where the piriformis, gemellus superior, obturator internus and part of quadratus femoris muscles were left intact, which combined with a standard capsule repair led to a reduced incidence of dislocation.[6]

In 2016, the Hip Unit at the Royal Devon and Exeter NHS Foundation Trust developed a modified technique using a posterior approach for hip hemiarthroplasties applicable to all patients. This approach involves division of only the obturator externus tendon and part of quadratus femoris muscle from their femoral insertions. These are repaired, along with the posterior capsule, at the end of the surgery, with strong non-absorbable suture (as opposed to a standard repair) through an enhanced trans-osseous technique onto the posterior aspect of the greater trochanter. This is a modification of the more extensive posterior repair technique developed by Carlton Savory, MD at The Hughston Clinic, Georgia, USA. The tendon insertions of piriformis, gemellus superior, obturator internus and gemellus inferior muscles are spared, and the extensive abductor muscle insertions of gluteus medius and gluteus minimus onto the greater trochanter are left undisturbed, minimising the potential negative impact of dividing muscles during surgery on postoperative recovery and mobility. This technique is named 'SPAIRE' as it allows the surgeon to 'Save Piriformis and Internus, Repairing Externus'. These muscles have been shown to act as the main extensor and abductor of the flexed hip which is of significant importance for movements such as rising from a chair or climbing stairs.[7] This contrasts to the standard lateral approach, where a significant proportion of the gluteal muscle insertions is divided, potentially impacting hip function. The combination of this muscle sparing approach with an enhanced capsule repair aims to provide sufficient stability to enable patients to mobilise full-weight bearing, without any of the specific restrictions currently included in routine postoperative posterior approach protocols. The preservation of these muscle insertions may replicate or even surpass the improved patient-related outcome measures observed in the study by Kristensen et al[5] and might achieve other benefits relating to more complete rehabilitation, reduced hospital stay and diminished requirement for social service support on discharge, with consequent savings from the health and social care budget.

## Review of existing evidence

A scoping review identified 13 studies that compared the traditional posterior approach with the lateral approach, including two randomised controlled trials (RCTs).[8] These evidence a trend for increased incidence of dislocation using the traditional posterior approach versus other approaches, with the exception of one paper describing good results with a modified version of the posterior approach.[9] However, this approach was only attempted in a small subset of the population prone to these fractures. Evidence with regard to other outcomes is inconsistent and of limited quality, with few RCTs to inform guidelines.[8]

## Aims

HemiSPAIRE is an RCT which, with its focus on postoperative function and mobility, aims to provide high quality evidence on the relative benefits of the SPAIRE surgical technique when compared with standard lateral technique. The findings may help to inform and update current guidelines.[4] In doing so it aims to contribute to improved function, mobility and quality of life outcomes for hip fracture patients, many of whom are elderly and frail, and reduce length of hospital stays.

## Primary objective

To test whether the SPAIRE technique improves postoperative function and mobility at 120 days following surgery in adults with a displaced intracapsular hip fracture requiring hemiarthroplasty, compared with the standard lateral approach through conducting a definitive two-arm RCT.

## Secondary objectives

Test whether the SPAIRE technique results in improved early function, mobility, pain and quality of life at 120 days, with reduced length of hospital stay, complication rates and mortality compared with the standard lateral approach, through collecting secondary outcome measures in the trial.

Investigate how patients experience the recovery period and mechanisms of recovery after surgery, by conducting a qualitative study with a sub-sample of patients in each trial arm.

Engage the contribution of a patient and public involvement (PPI) group to ensure the conduct and outputs of the study are relevant and useful to patients.

## METHODS AND ANALYSIS

### Trial design

This is a definitive pragmatic, multi-centre RCT in patients attending hospital with a displaced intracapsular fracture requiring hip hemiarthroplasty. Patients will be randomised to have their operation performed either by posterior approach (SPAIRE technique) or the standard lateral approach. Patients, ward staff and all research staff involved in postoperative evaluations will remain blinded to allocation. Outcomes will be recorded on the third postoperative day (POD3) and 120 days after surgery (POD120). These include measures of function, mobility, quality of life, pain, surgical complications, hospital length of stay, mortality, discharge destination and place of residence. Figure 1 shows a flow chart of the trial design.

### Trial setting

Six acute hospitals in the South West of England are recruitment centres, with a minimum of two surgeons per site. Training has been provided by the chief investigator (CI) and co-applicant JT, using lectures, mentorship,

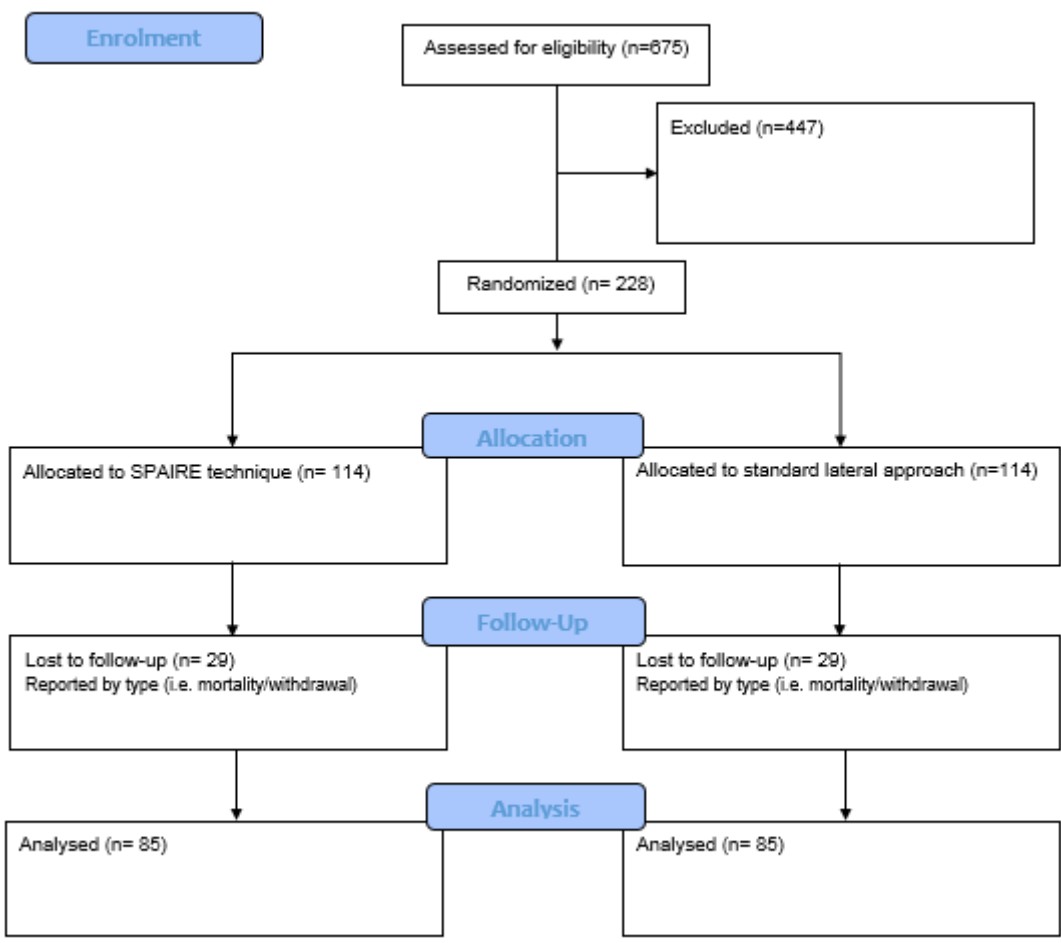

**Figure 1** Flow chart of trial design, with anticipated numbers of participants. SPAIRE, Save Piriformis and Internus, Repairing Externus.

observations, videos and one-to-one sessions as necessary. All participating surgeons have carried out a minimum of five cases using the posterior approach (SPAIRE technique) prior to participating in the study, and are prepared to undertake both procedures. Each participating surgeon uses both techniques. If a surgeon is not in equipoise, then he/she will not participate. Trainees who complete training under direct supervision of the principal investigator (PI) at each site are eligible to operate under supervision and are included on the list of surgeon collaborators.

## Participant eligibility criteria

All patients requiring hemiarthroplasty for a displaced intracapsular hip fracture are considered for inclusion. Inclusion criteria: patients aged 60 years or older presenting with an intracapsular hip fracture requiring hip hemiarthroplasty[4] who are resident in the South West of England. Exclusion criteria: patients who were immobile (unable to walk) before hip fracture, patients not expected to live until POD120 due to chronic illness and receiving surgery for palliative care, and use of femoral stems not of a proven stem design, in line with recommendations set by NICE clinical guideline on hip fracture management.[4] A sizeable proportion of this population suffer dementia and/or temporary delirium, and are not excluded; cognitive ability is not part of the eligibility criteria.

## Trial procedures

### Recruitment and participant identification

When a patient attends hospital with a confirmed intracapsular hip fracture, they are considered for inclusion. If potentially eligible the study is discussed with the patient and/or their carer(s). Surgery usually takes place 12 hours or more after admission or on the day after, providing patients with time to consider their participation. Potential participants are identified by orthopaedic surgeons involved in the study (co-applicants and collaborators) who admit patients under their care.

### Consent

Identified patients interested in participating in the study are invited to read the participant information sheet and, if interested, to provide informed consent. Researchers will be present to encourage potential participants to ask questions and to ensure participants fully understand the purpose of the research, risks associated with the intervention, obligations of participation and their right to withdraw at any time. The main consent forms must be fully completed and signed before patients are accepted on to the study (see online supplemental material provided). The qualitative consent form, if applicable, must be fully completed and signed between baseline and POD3. If a patient lacks capacity and is unable to consent, potential participation is discussed and consent later sought with a person whose relationship to the patient makes them suitable to act as his/her legal representative, for example, a

personal or professional consultee. If possible, the study is discussed or communicated to the participant in a way appropriate to their understanding. Participants who lack the mental capacity to consent and do not seem in agreement with any part of the study, even if agreement has been given by another, are not included. The participant's general practitioner is notified by letter that his/her patient is participating in the study.

## Randomisation scheme

Randomisation is undertaken as late as practically possible. There is no special preparation required in theatre, and no difference in equipment required for either surgical technique. Participants will be individually randomised to receive the SPAIRE or lateral procedure in a 1:1 ratio. Concealed allocation is determined by the UKCRC registered Exeter Clinical Trials Unit (CTU) using a validated password-protected web-based system. Allocation is based on random permuted blocks of varying size, and stratified by study site and by cognition level (impaired vs non-impaired). The surgeon is informed of allocation by the CTU via email through nhs.net mail. The CI and PI at site are copied in.

## Blinding

Patients, and research staff performing outcome assessments are blinded to treatment allocation. Surgeons and operative team are unblinded. An unblinded coordinator transcribes the surgical data to the database. There is no difference between the SPAIRE and lateral approach techniques in the following: surgical time taken, application of surgical dressing or postoperative care. For medico-legal reasons, the surgical approach used in the procedure is specified in the operation notes. A cover sheet is attached in front of the printed operation note stating that the patient is a participant in the trial and reminding the research team to avoid inadvertent unblinding to treatment allocation during postoperative assessments.

## Trial treatments

### Trial interventions

The two groups differ only in the surgical approach to the hemiarthroplasty. All preparation for surgery, patient positioning, skin incision, other aspects of surgery, surgical dressing and postoperative care, are the same according to current practice.

### SPAIRE technique through the posterior approach to the hip

If randomised to this arm of the trial, the SPAIRE technique is performed. This involves a modified muscle-sparing posterior approach where insertions of piriformis, superior gemellus, obturator internus and inferior gemellus are spared with division of only obturator externus and part of quadratus femoris. The single divided tendon and posterior capsule are subsequently repaired with a transosseous repair to their initial position prior to closure. The insertions of the abductor muscles are left intact throughout the procedure.

**Table 1** Trial outcome assessments, with time-points

| Objective | Outcome assessment | Tests conducted at |
| --- | --- | --- |
| **Primary** | | |
| Assess postoperative function and mobility | The Oxford Hip Score (OHS) | 120 days after surgery |
| **Secondary** | | |
| Assess postoperative function and early mobility | The De Morton Mobility Index (DEMMI) test | 3 and 120 days after surgery |
| | The Cumulated Ambulation Score (CAS) | 3 days after surgery |
| Assess pain | Numeric Pain Rating Scale (NPRS) | 3 and 120 days after surgery |
| Measure quality of life | EuroQol EQ-5D-5L | 120 days after surgery |
| Assess analgesia medication use | Analgesia medication use recorded from patient notes | 3 and 120 days after surgery |
| Measure hospital stay | Acute and total length of hospital stay | At time of discharge from acute and overall hospital stay |
| Assess any negative consequences of care | Adverse events (AE) and serious adverse events (SAEs), defined as any negative consequence of care resulting in unintended injury or illness | Continually throughout study participation, on average 120 days |
| Assess complication rates | Specific hip-related complications: dislocation, nerve injury, periprosthetic fracture, infection within 120 days of operation and need for re-operation (with reasons) | Within 120 days after surgery, recording the date of complication |
| Record discharge destination | The percentage of participants who are discharged to their prefracture place of residence | At time of discharge from hospital admission |
| Return to place of residence | Place of residence. The percentage of participants returning to their prefracture place of residence | At 120 days after surgery |
| Estimate survival | Mortality within 120 days of the operation | Within 120 days after surgery, recording the date of death |

### Lateral approach to the hip

If randomised to this arm of the trial, the patient is prepared and the operation performed in accordance with criteria set by the study, so as to minimise issues of standardisation with this approach. This means that the gluteus medius and minimus insertions onto the greater trochanter are partially divided anteriorly, leaving the posterior part of their insertions intact. The anterior capsule is divided and subsequently repaired prior to closure, followed by repair of the detached portion of the gluteal muscles.

### Baseline data collection and trial outcome assessments

Prefracture (baseline) participant characteristics, collected on the day of surgery, include: physical status at time of operation, measured using the American Society of Anaesthesiologists' (ASA) Physical Status Classification and level of cognition, determined by the surgical team, using the Abbreviated Mental Test. Prefracture mobility and quality of life measures, collected retrospectively at 3 days after surgery, include: Oxford Hip Score (OHS) and EuroQol EQ-5D-5L.

For details of trial outcome assessments see table 1.

### Qualitative research

We are conducting up to 20 semi-structured telephone interviews with patients (10 per arm) to examine their experience of the impact of surgery and recovery period, including factors such as pain, mobility, function, independence and quality of life. Participants are sampled from across participating sites. Interviews are conducted at POD120, after the quantitative data are collected, to gather information on patient experiences over the 4 months after their surgery. Written consent to take part in a telephone interview is included within the main trial consent form and confirmed verbally with patients when contacted for interview. Patients who lack capacity to consent are not contacted. The interview schedule is designed with advice from the patient and public involvement group, and two physiotherapists. Data will be analysed using thematic analysis,[10] with NVivo V.12.[11] The analysis will employ a combined deductive and inductive approach, and be underpinned by a critical realist perspective. We will triangulate the qualitative findings

with quantitative findings on mobility, function and quality of life.

## Patient safety and reporting

This trial follows Sponsor (Royal Devon and Exeter NHS Trust) standard operating procedures (SOPs) on safety reporting. Serious Adverse Events (SAEs) within this study are defined as any untoward medical occurrence that:

► Results in death.
► Is life-threatening.
► Requires inpatient hospitalisation or prolongation of existing hospitalisation.
► Results in persistent or significant disability/incapacity.
► Other 'important medical events' may also be considered serious if they jeopardise the participant or require an intervention to prevent one of the above consequences.

All SAEs will be reported to Exeter CTU and the local site Research and Development team within 24 hours of the PI being aware of the event. Exeter CTU follows up all SAEs. An SAE occurring to a participant is reported to the research ethics committee (REC) where, in the opinion of the CI the event was 'related' (resulted from administration of any of the research procedures) and 'unexpected' in relation to those procedures, within 15 working days of the CI becoming aware of the event. Any adverse event that does not fit the definition of serious above is recorded and reported in 6 monthly safety reports to the Independent Data Monitoring Committee/Trial Steering Committee (IDMC/TSC).

## Sample size

The primary outcome is the OHS measured at POD120. This gives a score between 0 and 48.[12 13] The minimal clinically important difference for OHS when comparing two groups has been estimated to be 5 points, with an SD of 9 points.[14] If we conservatively use an SD of 10 (ie, an effect size of 0.5), with 90% power, based on a t-test of two independent means at the 5% level of significance, we require 85 patients per trial arm, that is, a total of 170. Allowing for 25% drop-out, the total recruitment target is 228 (114 in each arm). On average, each participating site is expected to recruit around 38 patients in total, or around two to three patients per month, over 18 months.

## Monitoring recruitment and follow-up: internal pilot phase

Each site is expected to recruit two to three participants per month, but is allowed to recruit more participants if they are able to. The total target number recruited at 6 months is 72 participants. If the total number of participants at 6 months is 60 or more, the study will continue as planned. If the total number of participants at 6 months is less than 30, the trial will be stopped. If the number is between 30 and 60 participants, we will review procedures to see what improvements might be made, and discuss progress with the IDMC/TSC and the funder. Follow-up rates will also be reviewed at 6 months and if below what

has been predicted (ie, 75%) we will review procedures and see what improvements might be made. If recruitment is progressing as expected, but follow-up rates are not as expected at 10 months the IDMC/TSC will be asked to decide if the trial should continue.

## Statistical analysis

The primary analyses are pre-specified and a statistical analysis plan will be drafted and agreed by the IDMC/TSC and signed off by the independent statistician on the IDMC/TSC, prior to analyses. The study will be reported in accordance with the principles of Consolidated Standards of Reporting Trials guidelines.[15] There are no planned interim analyses. Primary analyses will be conducted on an intention-to-treat basis, blinded to group allocation.

The primary outcome, OHS at POD120, will be compared between study arms using linear regression, adjusting for site, cognitive impairment and prefracture characteristics (age, gender, place of residence, comorbidities (ASA grouped into categories: 1 or 2, 3, 4+)). Continuous secondary outcomes (De Morton Mobility Index (DEMMI),[16] Cumulative Ambulatory Score (CAS) and Numeric Pain Rating Scale (NPRS) at POD3; DEMMI, EQ-5D-5L, NPRS at POD120; acute and total length of stay) will be compared between study arms using linear regression, using the same adjustment variables. Results will be presented as means and SD in the two study arms and estimated mean differences with 95% CIs and p values. As we expect length of stay to be skewed, we will check the validity of the CIs for that outcome using bootstrap methods. Frequencies of death and surgical complications (by type) within 120 days follow-up will be presented, in the two study arms. If there is no clear evidence of non-proportional hazards, Cox regression will be used to analyse time to death, and time to complication. Discharge destination (ie, whether the same as prefracture place of residence), place of residence at 120 days (whether the same as prefracture place of residence) will be compared between study arms using logistic regression. For each of these outcomes, the number of events will be checked when considering adjustment factors to be included in the analyses.

Unadjusted analyses of the primary outcome and secondary outcomes will also be run.

Analysis of safety outcomes (including operative complications) will be based on the per-protocol population as well as on an intention-to-treat basis.

## Data management and confidentiality
### Data collection tools and source document identification

The data collection tool for this study is paper Case Report Forms (CRFs). Data are entered directly onto the CRFs and considered source documents. The CRF consists of standardised outcome measures (listed in table 1).

## Data handling and record keeping

The recruitment sites store all original signed informed consent forms and copies of the CRF pages. Information on these documents is transcribed at site to a trial database. The CTU Trial Manager checks the trial database for data completeness and liaises with sites regarding any data queries.

## Access to data

Direct access is granted to authorised representatives from the Sponsor, host institution and the regulatory authorities to permit trial-related monitoring, audits and inspections in line with participant consent.

## Data protection and patient confidentiality

The trial ID is used to identify data collected on CRFs and stored on the CTU database. Access to the CTU database is password protected and limited to those individuals necessary for quality control, audit and analyses. The Sponsor acts as the data controller for this study and will archive identifiable information for up to 5 years after the study has finished. Non-identifiable information will be kept in an open access archive managed by Exeter University indefinitely.

## Archiving

Archiving is authorised by the Sponsor following submission of the end of trial report. The Sponsor is responsible for archiving the essential documents. Exeter CTU is responsible for archiving the trial database. All essential documents will be archived for 5 years after completion of trial. Destruction of essential documents will require authorisation from the Sponsor.

## Monitoring, audit and inspection

A Trial Monitoring Plan has been developed and agreed by the Trial Management Group (TMG), IDMC/TSC (for charter contact lead author) and CI based on the trial risk assessment. Monitoring is being conducted by the CTU Trial Manager both remotely using the trial database and also with in person visits to sites. The CTU Trial Manager monitors participant enrolment, consent, eligibility and allocation to trial groups; adherence to trial interventions and policies to protect participants, including reporting of harm and completeness, accuracy and timeliness of data collection. Site staff are expected to assist the Trial Manager when requests for information are made or when an in person site visit is arranged.

## Patient and public involvement

The premise and the primary outcome of this trial was informed by discussions with patients about key outcomes for a separate project. We continue to integrate the involvement of patients in this work, organised by patient co-applicant AA and PPI facilitator EC. To date patients/carers with experience of hip fractures have helped develop the study design, plain English summary, patient facing documents and the qualitative interview schedule; they will also be closely involved in codesigning patient friendly dissemination materials.

## ETHICS AND DISSEMINATION
### Ethics

The protocol has been approved by the Yorkshire and the Humber—Bradford Leeds Research Ethics Committee (REC Reference: IRAS 258327). The trial is conducted in accordance with the study protocol, the principles of the Declaration of Helsinki, International Conference on Harmonisation of Good Clinical Practice and the Medicines for Human Use (Clinical Trials) Regulations, 2004.[17] Also in accordance with the UK Policy Framework for Health and Social Care Research[18] the Mental Capacity Act 2005[19] and the Data Protection Act 2018.[20] The trial has been adopted by the National Institute for Health Research (NIHR) Clinical Research Network and has relevant local NHS research approvals. The trial is sponsored by Royal Devon and Exeter NHS Trust, and managed by the UKCRC-registered Exeter CTU.

### Amendments

Sponsor SOPs are being followed for amendments. The decision to make an amendment is made by the TMG with Sponsor approval. The Sponsor decides whether an amendment is substantial or non-substantial. The Exeter CTU Trial Manager submits amendments to the REC, records approvals and communicates approved amendments to sites. Amendment history will be tracked and recorded in the Trial Master File maintained by Exeter CTU.

### Dissemination and impact activities

Trial progress is reported at TMG and IDMC/TSC meetings. We will follow established practice in our institution in disseminating the results of the HemiSPAIRE trial using the widest range possible of peer reviewed scientific journals, professional publications and national academic meetings. We will present at national and international conferences. If proven to be superior, the SPAIRE technique is likely to continue at recruiting sites and also adopted in other centres around the UK. Results will be incorporated into our clinical training programmes and we will make recommendations to regulatory bodies such as NICE. At the end of the trial, we will seek input from our PPI representatives to help disseminate a lay summary of the findings to study participants. Results will be disseminated via public involvement events, patient groups, networks and media releases.

**Author affiliations**
[1]Clinical Trials Unit, University of Exeter Medical School, Exeter, UK
[2]NIHR Applied Research Collaboration South West Peninsula (PenARC), University of Exeter Medical School, Exeter, UK
[3]Physiotherapy, Royal Devon and Exeter NHS Foundation Trust, Exeter, UK

[4]Research, Royal Devon and Exeter NHS Foundation Trust, Exeter, UK
[5]NIHR Applied Research Collaboration South West Peninsula (PenARC) patient engagement group, University of Exeter Medical School, Exeter, UK
[6]Psychology Applied to Health (PAtH) Group, Institute of Health Research, University of Exeter Medical School, Exeter, UK
[7]Research Design Service, Royal Devon and Exeter NHS Foundation Trust, Exeter, UK
[8]Exeter Hip Unit, Princess Elizabeth Orthopaedic Centre, Royal Devon and Exeter NHS Foundation Trust, Exeter, UK

**Acknowledgements** We would like to thank all those who have contributed to this study, including; the patients and carers who have been involved in the conception and planning of this research; colleagues for their advice and support. Also, the PIs, surgeon collaborators and research teams across the NHS Trust sites that have been involved in this study: Royal Devon and Exeter, Royal Cornwall Hospitals, Northern Devon Healthcare, Somerset, Yeovil Hospital and Weston Area Health.

**Contributors** The research idea was inspired by hip hemiarthroplasty patients. All co-applicants and external advisors actively contributed to the study design. SR, SB and JC developed the protocol. AP drafted the protocol ready for publication, manages the trial, and wrote amendments for ethics submission. JC leads the trial. RW is responsible for cross-site training on outcome measures and EG is lead nurse for the lead site. SB is the trial statistician. AA leads on patient and public involvement (PPI) representation. EC provides PPI expertise. JT provides surgical research expertise. RP reviewed the manuscript: he has been an RfPB-funded trial CI and has statistical skills, design and ethics committee experience. SM-T provides qualitative research expertise, supported by EC. SR provides senior trial management expertise, supported by AP as trial manager. All authors commented on the protocol and the manuscript.

**Funding** This study is funded by the National Institute for Health Research (NIHR) Research for Patient Benefit (PB-PG-0817-20039). The views expressed are those of the author(s) and not necessarily those of the NIHR or the Department of Health and Social Care. SB and EC are supported by the National Institute for Health Research Applied Research Collaboration South West Peninsula. The views expressed in this publication are those of the authors and not necessarily those of the National Institute for Health Research or the Department of Health and Social Care.

**Competing interests** AP, SB, EG, JT, AA, RW, RP, SM-T and EC have nothing to disclose. SR reports grants from NIHR RFPB, during the conduct of the study. JC reports grants from NIHR, during the conduct of the study.

**Patient consent for publication** Not required.

**Provenance and peer review** Not commissioned; externally peer reviewed.

**ORCID iDs**
Anna Price http://orcid.org/0000-0001-9147-1876
Susan Ball http://orcid.org/0000-0002-9937-4832
Sarah Morgan-Trimmer http://orcid.org/0000-0001-5226-9595

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
