## [Reviewer comments · BMJ Open]

ARTICLE DETAILS

TITLE (PROVISIONAL)	The effects of a modified muscle sparing posterior technique in hip hemiarthroplasty for displaced intracapsular fractures on post-operative function compared to a standard lateral approach (HemiSPAIRE); protocol for a randomised controlled trial.
AUTHORS	Price, Anna; Ball, Susan; Rhodes, Shelley; Wickins, Robert; Gordon, Elizabeth; Aylward, Alex; Cockcroft, Emma; Morgan-Trimmer, Sarah; Powell, Roy; Timperley, John; Charity, John

VERSION 1 – REVIEW

REVIEWER	Wolf, Olof Uppsala University, Orthopaedics
REVIEW RETURNED	03-Dec-2020

GENERAL COMMENTS	Dear authors, thank you for the opportunity to review your excellent study protocol. I have a few questions and remarks: I notice that the study started a year ago. So what is the study status? Is this protocol published half way through the study? Why so? It looks like the study is projected to finish inclusion in one year. I would suggest you state the eligibility for a hemiarthroplasty in lines 16-22 instead of stating that patients not eligible for a THA are eligible for this trial. One sentence with the criteria that make you consider a semi would be enough. The blinding of the ward staff worries me a bit. The sticker in front of the op note - do you think it is enough? Is it really important for the ward staff to be blinded? The post regime is the same any way. Leaving the assessors and the research staff blinded is important.
---

REVIEWER	Logishetty, Kartik Imperial College London
REVIEW RETURNED	01-Jan-2021

GENERAL COMMENTS	Thank you for the opportunity to review this well-designed RCT of a modified posterior approach vs. conventional lateral approach to treat intracapsular hip fractures requiring hemiarthroplasty. Introduction sufficiently describes clinical picture and interventions and equipoise. Methodology is robust with comprehensive SAP. I have five queries. I wonder if the OHS is discriminatory at POD3 and necessary as a secondary outcome measure. Secondly, it may be beneficial to describe the seniority of the surgeons eligible to operate in a little
---

	more detail. Hemiarthroplasty may be performed by trainees typically using a lateral approach. The SPAIRE is a modified and technically more difficult variation of the posterior approach, and thus facile for 'hip surgeons'. Therefore, are trainees eligible to operate for example under supervision of these trained experts? Third, please clarify the description of the "enhanced" capsular repair in the introduction. If this is the transosseous repair, a reference may assist the reader, E.g., Sioen et al, 2002 JBJSAm. Fourth, please clarify inclusion criteria. From my understanding, patients who could not provide informed consent are ineligible. However, the SAP describes analyses based on impaired vs. Unimpaired cognition. Was there a threshold on AMTS, for example? Finally, in line with the Review checklist, the authors have not discussed possible limitations of this study. Spirit checklist was well adhered to and the trial has been registered. Thank you, and best of luck conducting this RCT.
--	--

VERSION 1 – AUTHOR RESPONSE

Review Comment	Response	Details of changes made
Reviewer: 1 Dr. Olof Wolf, Uppsala University		
Comments to the Author:		
Dear authors, thank you for the opportunity to review your excellent study protocol. I have a few questions and remarks:		
I notice that the study started a year ago. So what is the study status? Is this protocol published half way through the study? Why so? It looks like the study is projected to finish inclusion in one year.	Thank you for your question. The protocol was written and approved by Ethics prior to the commencement of the study. The trial protocol was also registered on the Clinical Trials Register prior to commencement of recruitment. This submitted manuscript was written and submitted based on these documents. However, as this study included a 6 month internal pilot phase, the final draft of this paper was not	

	submitted for publication until the end of this phase (in case of change). Following the Pilot and due to Covid-19 some small changes were made to the Protocol. These were approved by our Independent Data Monitoring Committee, Sponsor, and Ethics committee (Protocol V2 2020-06-17). These changes are reflected in the submitted manuscript. Footnote: more recently, another minor change was made to the protocol with appropriate approvals (Protocol V3 2020-10-13). This was designed to facilitate recruitment of participants to the qualitative study. Following this opportunity for revisions to this manuscript this has been reflected in the final version that we are re-submitting.	P7, para 3: “Consent forms” changed to “The main consent forms” New sentence added “The qualitative consent form, if applicable, must be fully completed and signed between baseline and POD3.”
I would suggest you state the eligibility for a hemiarthroplasty in lines 16-22 instead of stating that patients not eligible for a THA are eligible for this trial. One sentence with the criteria that make you consider a semi would be enough.	Wording has been adjusted to be more specific, whilst still reflecting NICE guideline CG124: “NICE guidance CG124 on surgical procedures 1.6.2. Offer replacement arthroplasty (total hip replacement or hemiarthroplasty) to patients with a displaced intracapsular hip fracture. For patients who are not eligible for total hip replacement, hemiarthroplasty should be offered.”	P4, para2 Wording changed to: “The National Institute for Health and Care Excellence (NICE) recommends replacement arthroplasty (total hip replacement or hemiarthroplasty) to patients with a displaced intracapsular hip fracture. For patients who are not eligible for total hip replacement, hemiarthroplasty should be offered.”
The blinding of the ward staff worries me a bit. The sticker in	Thank you for this comment. We have clarified the text to	P8, para2

front of the op note - do you think it is enough? Is it really important for the ward staff to be blinded? The post regime is the same any way. Leaving the assessors and the research staff blinded is important.	communicate that blinding applies to Patients and Research Staff. While Ward staff are not aware of, or informed about, treatment allocations, it is not essential that they remain blinded. As you point out, the post treatment regime does not vary due to allocation in any case.	Blinding: The words "ward staff" have been deleted.
Reviewer: 2 Dr. Kartik Logishetty, Imperial College London		
Comments to the Author:		
Thank you for the opportunity to review this well-designed RCT of a modified posterior approach vs. conventional lateral approach to treat intracapsular hip fractures requiring hemiarthroplasty.	Thank you.	
Introduction sufficiently describes clinical picture and interventions and equipoise. Methodology is robust with comprehensive SAP.	Thank you.	
I have five queries.		
I wonder if the OHS is discriminatory at POD3 and necessary as a secondary outcome measure.	We are collecting OHS at POD3 where possible, but as it relates to the patient's mobility pre-fracture, it is not treated as an outcome measure. It will be summarised as a baseline characteristic in the two trial arms.	OHS at POD3 removed from Table 1 and added to text above Table 1 (pg8-9).
Secondly, it may be beneficial to describe the seniority of the surgeons eligible to operate in a little more detail. Hemiarthroplasty may be performed by trainees typically using a lateral approach. The SPAIRE is a modified and technically more difficult	Thank you for highlighting this. Wording has now been added to clarify the eligibility of trainees.	Pg6, para6; pg7, para1. Wording added for clarity: "Trainees who complete training under direct supervision of the principal investigator at each site are eligible to operate under supervision

variation of the posterior approach, and thus facile for 'hip surgeons'. Therefore, are trainees eligible to operate for example under supervision of these trained experts?		and are included on the list of surgeon collaborators.”
Third, please clarify the description of the "enhanced" capsular repair in the introduction. If this is the transosseous repair, a reference may assist the reader, E.g., Sioen et al, 2002 JBJSAm.	The SPAIRE technique has been developed by the Exeter Hip Unit (as referenced in the text). There is no formal reference for this technique available at present. We have added some descriptive details and the development process to the description in the text. Providing detailed information about and training for undertaking this repair will be part of the work that will be undertaken following delivery of this trial (dependent on the outcome). In the meantime readers are able to contact the corresponding author, or the Hip Unit at the Royal Devon and Exeter NHS Foundation Trust (as referenced in the text) for more information.	Pg5, para1. Additional details added to the description of the enhanced capsular repair. “(as opposed to a standard repair)... This is a modification of the more extensive posterior repair technique developed by Carlton Savory, MD at The Hughston Clinic, Georgia, USA... extensive... of gluteus medius and gluteus minimus onto the greater trochanter...” Please see text for full details.
Fourth, please clarify inclusion criteria. From my understanding, patients who could not provide informed consent are ineligible. However, the SAP describes analyses based on impaired vs. Unimpaired cognition. Was there a threshold on AMTS, for example?	Thank you for this question. Level of cognition is assessed by the surgical team, who make clinically informed decisions derived using the Abbreviated Mental Test (AMT). We had left these details out of the previous draft. Details of this assessment have now been added to Table 1, which will provide clarity. Cognition, or the ability for the patient to provide informed consent on their own behalf is not an exclusion criteria for this study. Please see Participant eligibility criteria Pg6, Para1. If a patient is judged to have impaired cognition, then a 'personal	Pg9, para 2 Baseline data collection and trial outcome assessments. The following words added to clarify: “Pre-fracture (baseline) participant characteristics, collected on the day of surgery, include: physical status at time of operation, measured using the American Society of Anaesthesiologists’ (ASA) Physical Status Classification and level of cognition, determined by the surgical team, using the Abbreviated Mental Test (AMT).”

	or professional consultee' can provide consent on their behalf. For more details, please see the Consent paragraph, Pg7, lines 8-10. We have now provided examples of the Participant consent and Consultee declaration forms as a supplementary file, which will provide additional clarity for the reader.	Pg7, para4. Wording added: "(see supplementary material provided)"
Finally, in line with the Review checklist, the authors have not discussed possible limitations of this study.	Is it possible that the 'Review checklist' mentioned is for articles which include a Discussion section, rather than for Protocol papers? Possible limitations of the study design are covered in Strengths and limitations of this study section, Pg3. We have added a bullet point outlining an additional potential limitation (high dropout rate due to vulnerable population) to this checklist. Editor, please advise if you would like further discussion of limitations in this submission.	Page3, para 1: Following bullet point added: "Due to the frail characteristics of the target population a relatively high dropout rate is anticipated, however this has been accounted for in the sample size calculation"
Spirit checklist was well adhered to and the trial has been registered.	Thank you.	
Thank you, and best of luck conducting this RCT.		

VERSION 2 – REVIEW

REVIEWER	Wolf, Olof Uppsala University, Orthopaedics
REVIEW RETURNED	05-Apr-2021
GENERAL COMMENTS	Thank you for the opportunity to review this revision of your protocol. My points of concern have all been addressed and I have no more comments. Good luck with finishing your RCT.

REVIEWER	Logishetty, Kartik Imperial College London
REVIEW RETURNED	01-Apr-2021
GENERAL COMMENTS	Thank you for addressing the queries raised in the first round of reviews. Good luck with the trial.